# Heregulin Drives Endocrine Resistance by Altering IL-8 Expression in ER-Positive Breast Cancer

**DOI:** 10.3390/ijms21207737

**Published:** 2020-10-19

**Authors:** Adriana Papadimitropoulou, Luciano Vellon, Ella Atlas, Travis Vander Steen, Elisabet Cuyàs, Sara Verdura, Ingrid Espinoza, Javier A. Menendez, Ruth Lupu

**Affiliations:** 1Center of Basic Research, Biomedical Research Foundation of the Academy of Athens, 115 27 Athens, Greece; adapapadim@gmail.com; 2Stem Cells Laboratory, Institute of Biology and Experimental Medicine (IBYME-CONICET), Buenos Aires C1428ADN, Argentina; luciano.vellon@ibyme.conicet.gov.ar; 3Environmental Health Science and Research Bureau, Health Canada, Ottawa, ON K1A 0K9, Canada; ella.atlas@canada.ca; 4Department of Biochemistry, Microbiology, and Immunology, University of Ottawa, Ottawa, ON K1N 6N5, Canada; 5Division of Experimental Pathology, Department of Laboratory Medicine and Pathology, Mayo Clinic, Rochester, MN 55905, USA; VanderSteen.Travis@mayo.edu; 6Program Against Cancer Therapeutic Resistance (ProCURE), Metabolism and Cancer Group, Catalan Institute of Oncology, 17007 Girona, Spain; ecuyas@idibgi.org (E.C.); sverdura@idibgi.org (S.V.); 7Girona Biomedical Research Institute (IDIBGI), 17190 Salt, Girona, Spain; 8Department of Preventive Medicine, John D. Bower School of Population Health, University of Mississippi Medical Center, Jackson, MS 39216, USA; iespinoza@umc.edu; 9Cancer Institute, School of Medicine, University of Mississippi Medical Center, Jackson, MS 39216, USA; 10Department of Biochemistry and Molecular Biology Laboratory, Mayo Clinic Minnesota, Rochester, MN 55905, USA; 11Mayo Clinic Cancer Center, Rochester, MN 55905, USA

**Keywords:** cytokines, IL-8, tamoxifen, autocrine, luminal

## Abstract

Sustained HER2/HER3 signaling due to the overproduction of the HER3 ligand heregulin (HRG) is proposed as a key contributor to endocrine resistance in estrogen receptor-positive (ER+) breast cancer. The molecular mechanisms linking HER2 transactivation by HRG-bound HER3 to the acquisition of a hormone-independent phenotype in ER+ breast cancer is, however, largely unknown. Here, we explored the possibility that autocrine HRG signaling drives cytokine-related endocrine resistance in ER+ breast cancer cells. We used human cytokine antibody arrays to semi-quantitatively measure the expression level of 60 cytokines and growth factors in the extracellular milieu of MCF-7 cells engineered to overexpress full-length HRGβ2 (MCF-7/HRG cells). Interleukin-8 (IL-8), a chemokine closely linked to ER inaction, emerged as one the most differentially expressed cytokines. Cytokine profiling using structural deletion mutants lacking both the *N*-terminus and the cytoplasmic-transmembrane region of HRGβ2—which is not secreted and cannot transactivate HER2—or lacking a nuclear localization signal at the *N*-terminus—which cannot localize at the nucleus but is actively secreted and transactivates HER2—revealed that the HRG-driven activation of IL-8 expression in ER+ cells required HRG secretion and transactivation of HER2 but not HRG nuclear localization. The functional blockade of IL-8 with a specific antibody inversely regulated ERα-driven transcriptional activation in endocrine-sensitive MCF-7 cells and endocrine-resistant MCF-7/HRG cells. Overall, these findings suggest that IL-8 participates in the HRG-driven endocrine resistance program in ER+/HER2- breast cancer and might illuminate a potential clinical setting for IL8- or CXCR1/2-neutralizing antibodies.

## 1. Introduction

The estrogen receptor (ER) is expressed in ~75% of breast carcinomas and is a key molecular driver and therapeutic target in these ER-positive (ER+) tumors [1]. Clinically, patients with ER+ breast cancer are treated with either selective ER modulators such as tamoxifen, which binds to and blocks ER, or selective estrogen degraders such as fulvestrant, which binds ER and induces its proteasomal degradation. Aromatase inhibitors, which lower the levels of estrogen by inhibiting the enzymatic conversion of androgens to estrogens, are typically employed as a second line of treatment in tamoxifen-resistant tumors. Despite the well-known ability of adjuvant endocrine therapy to prolong disease-free and overall survival in ER+ breast cancer, tumors in more than 50% of patients with metastases fail to respond, and nearly all metastatic and initially responsive tumors ultimately show tumor relapse due to acquired resistance [2,3,4,5,6,7]. One of the most predominant mechanisms of resistance to endocrine therapy in ER+ breast cancer is the cross-talk between ER and growth factor receptor pathways, which facilitates a transition from ligand (E_2_)-dependent to ligand-independent ER activation, ultimately leading to a reprogrammed ER transcriptional activity. Unfortunately, most of the intermediate molecular conduits linking growth factor receptor pathways to ER transcriptional reprogramming that promote endocrine-resistant breast cancer remain poorly understood.

Gene set enrichment analyses have revealed that ligand-induced transactivation of the human epidermal growth factor receptor 2 (HER2) or other members of the EGFR family in HER2-negative tumors generates an overlapping gene signature to that of the highly proliferative luminal-B subtype of ER+ breast carcinomas [8]. Indeed, HER-activating growth factors appear to contribute significantly to the endocrine-resistant phenotype, underpinning the poor clinical outcome of luminal-B tumors treated with tamoxifen. A clinical study examining a large retrospective group of tamoxifen-treated patients with ER+ breast cancer revealed that HER1+/HER3+ patients were significantly more likely to relapse on tamoxifen [9]. In addition, tamoxifen-sensitive luminal-like breast cancer cells exogenously treated with the HER3 ligand HRGβ1, or engineered to overproduce the HER3 ligand HRGβ2, have been shown to acquire a bona fide endocrine-resistant phenotype in terms of estrogen-independent growth and refractoriness to anti-estrogen therapies, both in vitro and in vivo [10,11,12,13,14,15]. Beyond the overlapping downstream activation of PI3K/AKT and MAPK signaling cascades present in ER+ breast cancer cells overexpressing HER2 [16,17,18], which also acquire an endocrine-resistant phenotype, the definitive mechanisms linking HER2 transactivation by HRG-bound HER3 to the acquisition of a hormone-independent phenotype in ER+ breast cancer cells are largely unknown.

Both the quantitative and qualitative repertoire of autocrine and paracrine cytokines are emerging as potential contributors to the metastatic and therapy-resistant phenotype of breast cancer in an ER-related manner. For instance, interleukin-8 (IL-8/ CXCL8), a prototypical member of a superfamily of small, inducible secreted CXC chemokines, is positively associated with invasiveness and angiogenic and metastatic potential in breast cancer cells and is negatively linked to ER status [19,20,21,22,23,24,25,26]. Here, we explored the possibility that autocrine HRG signaling drives cytokine-mediated endocrine resistance in ER+ breast cancer cells. Using cytokine antibody array technology [21,27] and structural HRG mutants with specifically altered capacities to be secreted and transactivate HER2 [28,29], we provide phenotypic and mechanistic evidence that IL-8 might operate as an enabling factor promoting estrogen-independency and anti-estrogen resistance in HRG-overexpressing ER-positive/HER2-negative breast carcinoma cells.

## 2. Results

### 2.1. Identification of IL-8 as an HRG-Driven Cytokine Using Chemokine Antibody Array Technology

We used human cytokine antibody array analysis to identify differential HRG-driven cytokine responses in ER+ breast cancer cells. Using the RayBio™ (Norcross, GA, USA) C-series (C7) Human Cytokine Array, we simultaneously screened the expression of 60 different cytokines, chemokines, growth factors, angiogenic factors, and proteases (Appendix A) in a conditioned medium from luminal A-like MCF-7 breast cancer cells engineered to overexpress either the full-length version of HRGβ2 [28] or structural deletion mutants lacking specific domains that drive the extracellular versus nuclear localization of HRGβ2 [29] (Figure 1A). Specifically, the HRG-M4 structural mutant lacks both the nuclear localization signal (NLS) at the *N*-terminus and the cytoplasmic-transmembrane region of HRG, thereby generating a cytoplasm-restricted version that cannot be secreted and cannot transactivate HER2. The HRG-M1 structural mutant exclusively lacks the NLS at the *N*-terminus, thereby impairing the ability of HRG to localize in the nuclei of breast cancer cells. In contrast to HRG-M4, HRG-M1 can be secreted into the extracellular milieu and can effectively induce transactivation phenomena with HER2 (Figure 1A).

Figure 1A also shows the raw data images from the cytokine antibody array using MCF-7/pBABE (control), MCF-7/HRG, MCF-7/HRG-M1, and MCF-7/HRG-M4 cells. Densitometric analyses suggested a slight elevation in the secretion of the urokinase-type plasminogen activator receptor (uPAR) and the EGFR (HER1) ligand amphiregulin in response to HRGβ2 overexpression in MCF-7/HRG cells. MCF-7/HRG cells further showed a noteworthy up-regulation of IL-8. MCF-7/HRG-M1 cells generated a similar cytokine profile to that of MCF-7/HRG cells, which was characterized by the conspicuous up-regulation of IL-8. By contrast, MCF-7/HRG-M4 cells failed to up-regulate IL-8, but did show an up-regulation of uPAR and amphiregulin secretion. Quantitative determination of IL-8 levels by enzyme-linked immunosorbent assay (ELISA) confirmed the semi-quantitative array data (Figure 1B). Specifically, MCF-7/pBABE control cells secreted 131 ± 14 pg IL-8 mg protein^−1^, whereas MCF-7/HRG, MCF-7/HRG-M4, MCF-7/HRG-M1 cells expressed 440 ± 10, 87 ± 14, and 472 ± 19 pg IL-8 mg^−1^, respectively.

### 2.2. HRG Overexpression in HER2-Negative Breast Cancer Cells Qualitatively Phenocopies the IL-8 Cytokine Signature Driven by her2 Overexpression

Using the antibody-based RayBio™ (Norcross, GA, USA) Human Cytokine Array III, which simultaneously detects 42 cytokines and growth factors on one membrane, we previously demonstrated that HER2 overexpression in MCF-7 cells robustly up-regulated the expression of IL-8 and the alpha-isotype of the growth-related oncogene (GRO; CXCL1) chemokine [27]. To test whether the HRG-driven cytokine signature was merely a phenocopy of that promoted by HER2 overexpression, we re-screened the conditioned medium of MCF-7/Her2-18 transfectants with the RayBio™ (Norcross, GA, USA) C-series (C7) Human Cytokine Array. MCF-7/Her2-18 cells overexpress full-length HER2 cDNA under the control of the SV40 promoter and accumulate ~45-times the level of HER2 protein of parental MCF-7 cells [16]. Similar to MCF-7/HRG cells, MCF-7/Her2-18 cells notably augmented the secretion of uPAR, amphiregulin and, particularly, IL-8, when compared with MCF-7/neo control counterparts (Figure 2A). In contrast to MCF-7/HRG cells, however, MCF-7/Her2-18 cells also showed an elevated secretion of TIMP-2, VEGF, and GRO relative to control cells. Although qualitatively similar in terms of IL-8 expression, when compared with MCF-7/pBABE and MCF-7/neo control cells, quantitative analysis of extracellular IL-8 levels by ELISA revealed a 12-fold increase in IL-8 secretion from MCF-7/Her2-18 cells, but only a 3.6-fold increase in MCF-7/HRG cells (Figure 2A).

### 2.3. HRG-Driven Regulation of IL-8 Is ER-Dependent

IL-8 is preferentially secreted in ER-negative breast cancer cells; indeed, no ER+ breast cancer cell line tested thus far has been found to express detectable levels of IL-8 [19,20,21]. Moreover, exogenous ER expression down-regulates IL-8 expression in ER-negative MDA-MB-231 cells, naturally overexpressing IL-8. Our finding that HRG overexpression in an ER+ background suffices to up-regulate IL-8 suggested the occurrence of a specific scenario of HRG-driven IL-8 overproduction in ER-positive/HER2-negative breast cancer cells. To test this, we employed MDA-MB-231 cells, a cancer cell line that naturally overexpresses HRGβ2 and IL-8, and a clonal derivative of this cell line stably transfected with an antisense HRGβ2 construct (MDA-MB-231/AS-31) [30], leading to almost undetectable HRGβ2 expression [30,31,32,33]. We thus characterized the cytokine signature of MDA-MB-231/AS-31 cells as an engineered model of ER-negative/HRG-negative breast cancer cells. Our results showed that whereas the blockade of HRG expression slightly decreased the amounts of secreted TIMP-2, uPAR, and VEGF in MDA-MB-231 cells, as determined with the RayBio™ (Norcross, GA, USA) C-series (C7) Human Cytokine Array, IL-8 overexpression become even more conspicuous in MDA-MB-231/AS-31 cells (Figure 2B). Specifically, ELISA determination of IL-8 levels confirmed a small but significant (35%) increase in secreted IL-8 in MDA-MB-231/AS-31 cells as compared with MDA-MB-231/AS-V (vector control) counterparts (Figure 2B).

### 2.4. Functional Blockade of IL-8 Regulates ER Transcriptional Activity in an HRG-Dependent Manner

To evaluate the effects of functional blockade of IL-8 on ERα-transactivation and E_2_ responsiveness, we co-transfected MCF-7/pBABE and MCF-7/HRG cells with a Luciferase reporter gene linked to an Estrogen Response Element (ERE-Luciferase) and also with the internal control vector pRL-CMV. Transfected cells were then evaluated for changes in the levels of basal (E_2_-independent) reporter activity in the presence of increasing concentrations of an anti-IL-8 antibody (Figure 3).

In the absence of E_2_, the basal transcriptional activity of the ER reporter was increased in a dose-dependent manner in MCF-7/pBABE cells by treatment with an anti-IL-8 antibody, relative to untreated control cells (Figure 3A, top). Specifically, ER-dependent luciferase activation was increased 3.7-, 6.3- and 13.1-fold in the presence of 1, 5, and 10 μg/mL of anti-IL-8 antibody, respectively. We then transiently transfected MCF-7/pBABE cells with the ERE-Luciferase construct as before and treated them with a combination of E_2_ and anti-IL-8 antibody to determine whether their actions were additive, synergistic, or antagonistic. As a single agent, E_2_ (10^−9^ mol/L) induced a ~5-fold increase in Luciferase activity relative to basal levels in untreated cells (Figure 3A, top). Remarkably, co-exposure of MCF-7/pBABE cells to E_2_ and anti-IL-8 antibody resulted in a dose-dependent increase (up to ~70-fold at 10 μg/mL) in ERE-reporter activity (Figure 3A, top), representing a ~12-fold increase in ERα-dependent transcriptional activity when compared to the activity in E_2_-stimulated MCF-7/pBABE cells. Accordingly, the concentration of anti-IL8 antibody to produce a half-maximal effect was notably lower in the presence of E_2_ (~1 μg/mL) than in its absence (~5 μg/mL).

A completely different picture emerged when the impact of IL-8 blockade on ER transcriptional activity was measured in MCF-7/HRG cells using the same protocol. In the absence of E_2_ stimulation, we failed to observe any significant ER activation and instead observed a slight inhibition of the ER transcriptional activity in response to the functional blockade of IL-8 (Figure 3A, bottom). Such dose-dependent inhibitory behavior of ER transcriptional activity observed at the higher doses of the anti-IL8 antibody was unaffected by the presence of E_2_. Accordingly, the dose of anti-IL8 antibody required to produce a 50% reduction of the maximum effect did not significantly vary between E_2_-independent (~5 μg/mL) and E_2_-stimulated (~4 μg/mL) conditions. Indeed, the ability of the anti-IL-8 antibody to decrease the E_2_-independent and E_2_-induced transcriptional activity of ERα was more noticeable when the fold-increase in the ERE activity of MCF-7/HRG cells was inter-normalized to the ERE activity in MCF-7/pBABE cells (Figure 3B).

## 3. Discussion

Here, we explored the possibility that autocrine HRG signaling drives cytokine-related endocrine resistance in ER+ breast cancer cells. Analysis of the secreted “cytokine signature” in the extracellular milieu of luminal-like ER+ breast cancer cells exhibiting an HRG-driven endocrine-resistant phenotype revealed the differential upregulation of three well-known regulators of ERα activity, namely, uPAR, amphiregulin, and IL-8. The transition in uPAR signaling from transient and uPA-dependent to sustained and autonomous provides a selective advantage for ER+ breast cancer cells in the absence of E_2_, thereby serving as an escape pathway for breast cancer cells from ERα-targeting therapeutics equivalent to that provided by HER2 overexpression [34]. The epidermal growth factor receptor (EGFR/HER1) ligand amphiregulin is a key effector of ERα activity. Whereas endocrine therapy induces suppression of amphiregulin in responsive cells, the activation of an amphiregulin/EGFR autocrine loop has been proposed to drive endocrine-resistant phenotypes in breast cancer [35,36,37]. The synthesis and secretion of IL-8 in breast cancer cells are known to closely relate to ER status [19,20,21,22,23,24,25,26]; IL-8 secretion is apparently low in ER-positive breast cancer cell lines and tends to be high in ER-negative cells. Autocrine production of IL-8 is inversely associated with cancer cell responses to tamoxifen, and tamoxifen-resistant derivative cells exhibit elevated IL-8 expression, thereby suggesting that IL-8 could also be a cause and indicator of endocrine resistance as a part of the IL-8-mediated program of metastatic progression of breast cancer [19,20,21,22,23,24,25,26]. Using structural mutants of HRG with altered capacities to distribute throughout intracellular versus extracellular compartments, we found that IL-8, but not uPAR or amphiregulin, was the sole trait of the HRG-driven “cytokine signature” involving autocrine signaling, leading to sustained transactivation of HER2.

HRG-driven activation of IL-8 expression in ER+ cells was found to require HRG secretion and transactivation of HER2, but not HRG nuclear localization. The occurrence of self-sustaining autocrine signaling of HRG might, therefore, suffice to establish a positive link between IL-8 expression, ERα activity, and endocrine therapy responsiveness independently of HER2 overexpression. Indeed, our findings might add a new dimension to the ongoing controversy regarding ER/IL-8 cross-talk, namely, the inverse correlation between IL-8 expression with ER status (i.e., ER downregulates IL-8 expression independently of estradiol) and the positive correlation between estradiol and IL-8 (i.e., estradiol seems to augment IL-8 expression). On the one hand, the ability of HRG to up-regulate IL-8 expression cannot be established in the absence of ER [19,20]. Accordingly, the IL-8-overexpressing phenotype of ER-negative MDA-MB-231 remained unaffected upon the silencing of HRG. On the other hand, IL-8 might affect the endocrine responsiveness of ER-positive breast cancer cells through modulation of ER activity. When we tested whether IL-8 could alter the transcriptional activity of Erα, we found that incubation of ER+/HRG-negative breast cancer cells with an antibody against IL-8 triggered a dramatic, dose-dependent increase in ER transcriptional activity in endocrine-responsive cells, which was more pronounced when combined with E_2_. It should be noted that blockade of IL-8 was sufficient to activate ERα transcriptional activity and that the magnitude of the activation by the combination of the anti-IL-8 antibody and E_2_ was far greater than the naturally saturated response of ligand (E_2_)-occupied ERα in low-IL-8-expressing breast cancer cells. These observations, together with the fact that the direction of the effects of IL-8 targeting on ERα transcriptional activity switched in an HRG-dependent manner, support a possible mechanistic framework where IL-8 might operate as a corepressor for ERα itself by directly controlling ERα activity or, alternatively, by activating one or several intermediary ERα corepressing mechanisms such as expression/activity of chromatin remodelers, competition with ERα coactivators, or alteration of ERα protein stability, among others [38,39,40,41,42,43,44,45,46].

HRG-induced formation of HER2/HER3 heterodimers—which are considered the most potent pairs with respect to the strength of interaction, ligand-induced tyrosine phosphorylation, and downstream signaling—will enhance IL-8 expression downstream of overactive MAPK and PI3K signaling pathways, whereas autocrine signaling of IL-8 via its CXCR1/2 receptor will enhance and prolong further HER2/HER3-mediated signaling [47,48,49]. Thus, in HRG-overexpressing ER+/HER2-negative cells, specific neutralization of IL-8 bioactivity likewise reduces the ability of HRG overexpression to increase the non-genomic (e.g., MAPK- and PI3K-induced) unliganded transcriptional activity of ERα. The finding that anti-IL-8 treatment reduced but did not completely block the E_2_-independent hyperactivity of ERα in the continuous presence of up-stream oncogenic stimuli such as HRG suggests a need not only for combined treatment with drugs capable of impeding ligand-induced HER2/HER3 signaling (e.g., pertuzumab) but also the potential involvement of additional HRG-driven ERα co-activating cytokines (e.g., IL6, which is known to be repressed by E_2_-driven activation of ERα but drives endocrine therapy resistance by various mechanisms including direct transcriptional activation of ERα [50,51,52,53,54,55,56]). In this complex scenario of multiple cytokines driving endocrine resistance in HRG-overexpressing ER+ breast cancer cells, the sole blockade of IL-8 bioactivity might become inadequate in the presence of an augmented availability of IL-6 downstream of the HRG-activated HER2/HER2 oncogenic unit. Nonetheless, HRG/HER2:HER3-induced autocrine secretion of IL-8 can be viewed as part of the endocrine resistance program in HRG-overexpressing ER+ breast cancer cells by controlling the magnitude of the estrogen response and mediating the anti-estrogen inhibition of ERα [57]. Therefore, it is, tempting to suggest that IL-8 might be part of a context (HRG)-dependent regulatory mechanism that dictates endocrine responsiveness in ER+/HER2-negative breast cancer cells (Figure 4). In the absence of sustained HER2:HER3 signaling, blockade of such negative feedback will promote an exacerbated ERα transcriptional activation. In the presence of persistent HRG-induced activation of HER2:HER3 signaling, the resulting augmentation of IL-8 secretion will further potentiate the non-genomic (e.g., MAPK- and PI3K-driven) potentiation of the unliganded transcriptional activity of ERα (e.g., via transactivation of HER2 [58,59]) characteristic of the endocrine-resistant phenotype in HRG-overexpressing ER+ breast cancer cells. Intriguingly, the “cytokine events” driven by sustained HRG-induced transactivation of HER2 via HER2/HER3 heterodimers were similar but not identical to those promoted by high levels of HER2 homodimers in HER2-overexpressing breast cancer cells. Whether the different signaling output dictated by the oligomeric conformations of HER2 in HRG-positive/HER2-negative and HRG-negative/HER2-positive breast cancer cells [60] might explain the partial overlap between their respective cytokine signatures will require further investigation.

It has been postulated that the net effect of IL-8 action in breast cancer progression involves a balance between the promotion of ER inaction and the occurrence of HER2 overexpression. Accordingly, IL-8 overexpression is characteristic of basal-like (ER-negative) and HER2-enriched (HER2+) intrinsic subtypes of breast carcinomas. However, the putative significance of IL-8 in ER-positive/HER2-negative breast carcinomas has remained largely unexplored. Because IL-8, through engagement with its receptors CXCR1/2, is a well-known driver of the stemness properties of the highly tumorigenic and resistant to cancer therapy sub-populations of so-called cancer stem cells [61,62,63], it is tempting to suggest that IL-8- or CXCR1/2-neutralizing antibodies might be therapeutically relevant for the clinical management of particular subsets of ER-positive/HER2-negative luminal breast carcinomas overexpressing the HER3 ligand HRG. In this regard, we do recognize that the main weakness of our study is the lack of experimental evidence supporting the antitumor activity of IL-8- or CXCR1/2-neutralizing antibodies in cultured and xenografted HRG-overexpressing/ER+ breast cancer cells. Furthermore, the therapeutic relevance of our findings certainly requires the analysis of breast cancer series to validate the driving role of HRG in endocrine therapy resistance. Only then will we be able to support the prognostic and therapeutic relevance of the HRG/IL-8/ER cross-talk in breast cancer patients.

## 4. Materials and Methods

### 4.1. Materials

Phenol red-free Improved Minimal Essential Medium (IMEM) was from Biofluids, Inc. (Rockville, MD, USA). Dextran-coated charcoal-treated bovine serum (CCS) was from Biosource International (Camarillo, CA, USA). RayBio^TM^ C-series (C7) Human Cytokine Array was purchased from RayBiotech, Inc. (Norcross, GA, USA). E_2_ was from Sigma-Chemical Co. (St. Louis, MO, USA). The anti-human CXCL8/IL-8 monoclonal antibody (clone 6217)–a functional antibody capable of neutralizing CXCL8 bioactivity was from R&D Systems, Inc. (Minneapolis, MN, USA; Cat. # MAB208).

### 4.2. Cell Lines and Culture Conditions

The PCR products generated using the HRGβ_2_ cDNA accession number 183996 (full-length HRGβ_2_) or the structural deletion mutants (HRGβ_2_-M4 and HRGβ_2_-M1) were cloned into the retroviral expression vector pBABE-Puro using *Bam*HI and *Eco*RI restriction sites. Retroviral constructs were transfected into a high efficiency transient amphotropic packaging system (TSA54 cell line) with FuGENE reagent. Retrovirus-containing medium collected after 48 h was used to infect MCF-7 cells for 24 h in the presence of Polybrene (Sigma-Chemicals, St. Louis, MO, USA). Infected MCF-7 cells were grown for an additional 24 h in standard medium and stable cell lines (MCF-7/pBABE, MCF-7/HRGβ_2_, MCF-7/HRGβ_2_-M4, and MCF-7/HRGβ_2_-M1) were selected and expanded in the presence of 2.5 µg/mL puromycin for 2 weeks. Expression levels of HRGβ_2_, HRGβ_2_-M4, and HRGβ_2_-M1 were assessed by RT-PCR using the Gene Amp Kit (Promega Corporation, Madison WI, USA). MCF-7 cells stably overexpressing HER2 (clone MCF-7/Her2-18), and their vector-transfected counterparts (MCF-7/neo) were kindly provided by Dr. Mien-Chie Hung (The University of Texas M.D. Anderson Cancer Center, Houston, TX, USA).

### 4.3. Conditioned Medium

To prepare a conditioned medium, cells were plated in 100 mm tissue-culture dishes until they reached 75–80% confluence. Cells were washed twice with serum-free IMEM and incubated overnight in serum-free IMEM. Cells were then cultured for 48 h in a low-serum (0.1%) medium. The supernatants were collected, centrifuged at 1,000 × *g*, aliquoted, and stored at −80 °C until testing.

### 4.4. Cytokine Antibody Arrays

Assays for cytokine antibody arrays were carried out as per the manufacturers’ instructions. Briefly, cytokine array membranes were blocked with 5% BSA/TBS (0.01 mol/L Tris-HCl pH 7.6/0.15 mol/L NaCl) for 1 h. Membranes were then incubated with ~2 mL of conditioned media prepared from the different cell lines after normalization for equal amounts of protein. After extensive washing with TBS/0.1% *v/v* Tween 20 (3 times, 5 min each) and TBS (2 times, 5 min each) to remove unbound material, the membranes were incubated with a cocktail of biotin-labeled antibodies against different individual cytokines. The membranes were then washed and incubated with horseradish peroxidase (HRP)-conjugated streptavidin (2.5 pg/mL) for 1 h at room temperature. Unbound HRP-streptavidin was washed out with TBS/0.1% *v/v* Tween 20 and TBS. Signals were finally detected using the ECL system (Amersham, Arlington Heights, IL, USA). Densitometric values were quantified using Scion Imaging Software (Scion Corp., Frederick, MD, USA).

### 4.5. IL-8 ELISA

IL-8 levels were measured using a quantitative immunometric sandwich ELISA following the procedures recommended by the manufacturer (R&D Systems, Minneapolis, MN, USA). Triplicate cultures were tested for each experimental condition.

### 4.6. ER Transcriptional Activity

Cells were propagated in E_2_-deprived (phenol red-free) IMEM with 5% CCS for 5 days before the onset of experiments. For experiments, the cells were seeded into 12-well plates at 1 × 10^5^ cells/well. Cells were transfected using the FuGENE 6 reagent (Roche Biochemicals, Indianapolis, IN, USA) with 1 μg/well of the estrogen-responsive reporter, ERE-Luc, containing a *Xenopus* vitellogenin A_2_-derived ERE, along with 0.1 μg/well of the internal control plasmid pRL-CMV, used to correct for transfection efficiency. After 18 h, cells were washed and then incubated in fresh medium containing 5% CCS, supplemented with E_2_ (10^−9^ mol/L), anti-IL-8 antibody (0.1, 0.5, 1, 5, and 10 μg/mL), combinations of these compounds as specified, and vehicles (*v/v*) alone. Approximately 24 h after treatments, Luciferase activity from cell extracts was measured using a Luciferase Assay System (Promega, Madison, WI, USA) on a TD-20/20 luminometer (Turner Designs, Sunnyvale, CA, USA). The magnitude of activation in ERE-Luciferase-transfected cells treated with the vehicle was determined after normalization to the activity of pRL-CMV and was defined as 1.0-fold. This control value was used to calculate the relative (fold) change in transcriptional activities of ERE-Luciferase-transfected cells in response to treatments after normalization to pRL-CMV activity. All data were normalized as the ratio of raw light units to pRL-CMV units corrected for pRL-CMV activity, and were shown as the mean ± S.D. from 3 separate experiments (performed in triplicate).

### 4.7. Statistical Analysis

For all experiments, at least 3 independent experiments were performed with *n*≥3 replicate samples per experiment. Investigators were blinded to animal data allocation. Experiments were not randomized. Data were presented as mean ± S.D. Comparisons of means of ≥3 groups were performed by one-way ANOVA and Dunnett’s t-test for multiple comparisons using GraphPad Prism (GraphPad Software, San Diego, CA, USA). In all studies, *p*-values <0.05 and <0.005 were considered to be statistically significant (denoted as * and **, respectively). All statistical tests were two-sided.

## 5. Conclusions

Persistent promotion of HER2/HER3 signaling due to the overproduction of the HER3 ligand HRG suffices to upregulate IL-8 expression as part of the endocrine-resistant phenotype in ER-positive/HER2-negative breast cancer cells. These findings might illuminate a potential clinical setting for IL8- or CXCR1/2-neutralizing antibodies in ER+ breast carcinomas overexpressing HRG.

## Figures and Tables

**Figure 1 ijms-21-07737-f001:**
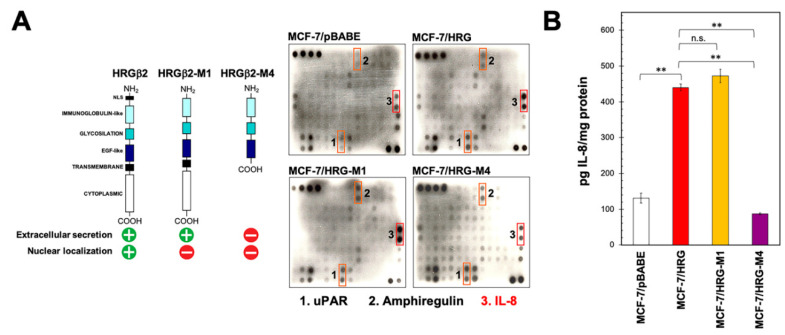
Autocrine heregulin up-regulates IL-8 in luminal breast cancer cells. (**A**) Left. Schematic representation of the structural heregulin (HRG) mutants employed. Right. Forty-eight-hour conditioned media from MCF-7/pBABE, MCF-7/HRG, MCF-7/HRG-M1, and MCF-7/HRG-M4 cells were assayed for cytokine content as described in the “Materials and methods” section. Shown are representative results (*n* = 3) revealing conspicuous changes in uPAR, amphiregulin, and IL-8 secreted from MCF-7/HRG cells as compared with MCF-7/pBABE control counterparts. (**B**) IL-8 concentration in conditioned media from MCF-7/pBABE, MCF-7/HRG, MCF-7/HRG-M1, and MCF-7/HRG-M4 cells was assessed by ELISA. Values represent mean (columns) ± S.D. (bars) from three independent experiments. (** *p* < 0.005; n.s. not statistically significant).

**Figure 2 ijms-21-07737-f002:**
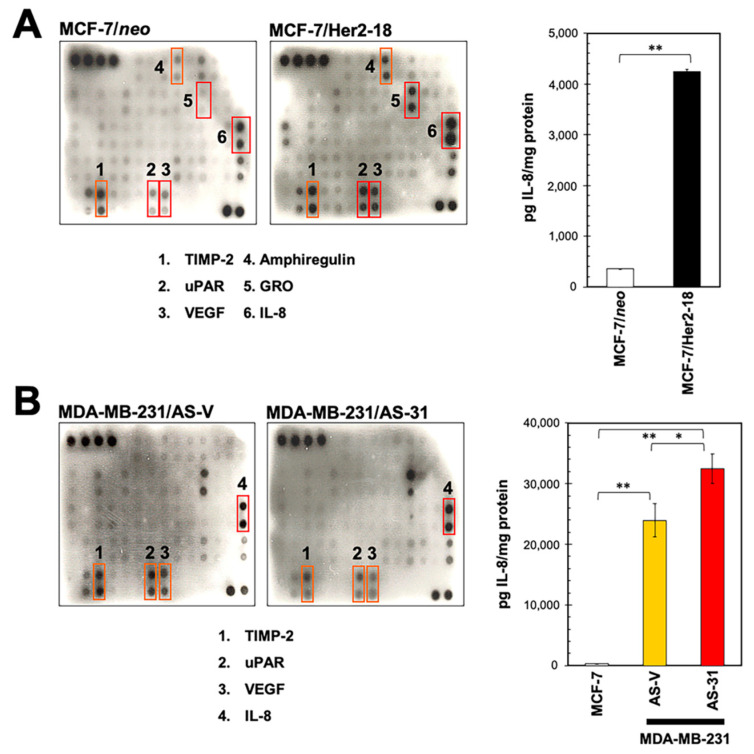
(**A**) HRG- and HER2-induced cytokine signatures are similar but not identical. Left. Forty-eight-hour conditioned media from MCF-7/neo and MCF-7/Her2-18 cells were assayed for cytokine content as described in the “Materials and methods” section. Shown are representative results (*n* = 3) revealing conspicuous changes in TIMP-2, uPAR, VEGF, amphiregulin, GRO, and IL-8 secreted from MCF-7/Her2-18 cells as compared with MCF-7/neo control counterparts. Right. IL-8 concentration in conditioned media from MCF-7/neo and MCF-7/Her2-18 cells was assessed by ELISA. Values represent mean (columns) ± S.D. (bars) from three independent experiments. (** *p* < 0.005). (**B**) Suppression of HRG overexpression is not sufficient to down-regulate IL-8 overexpression in ER-negative breast cancer cells. Forty-eight-hour conditioned media from HRG-/IL8-overexpressing MDA-MB-231/AS-V cells and the HRG-negative MDA-MB-231/AS-31 clone were assayed for cytokine content as described in the “Materials and methods” section. Shown are representative results (*n* = 3) revealing conspicuous changes in TIMP-2, uPAR, VEGF, and IL-8 secreted from MDA-MB-231/AS-31 cells as compared with MDA-MB-231/AS-V control counterparts. IL-8 concentration in conditioned media from MCF-7, MDA-MB-231/AS-V, and MDA-MB-231/AS-31 cells was assessed by ELISA. Values represent mean (columns) ± S.D. (bars) from three independent experiments. (* *p* < 0.05; ** *p* < 0.005).

**Figure 3 ijms-21-07737-f003:**
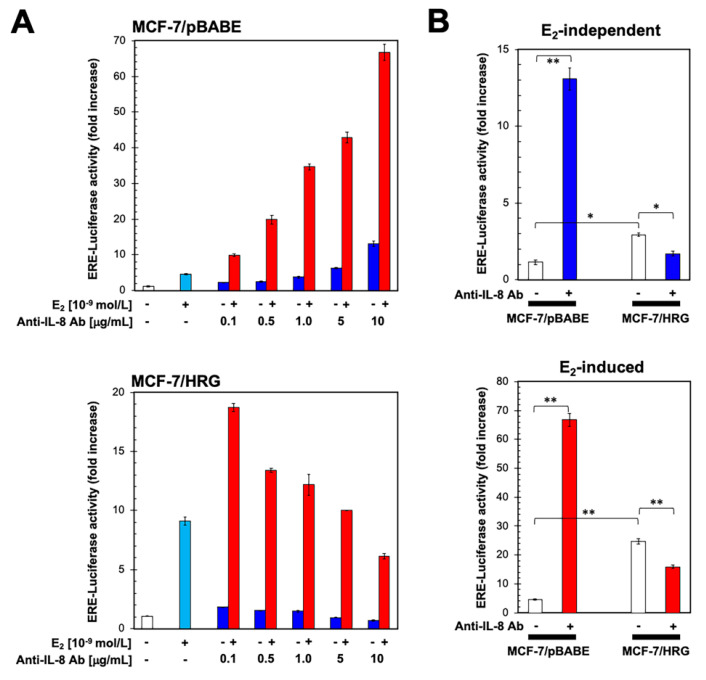
Blockade of IL-8 bioactivity differentially regulates ERα-driven transcriptional activity in an HRG-dependent manner. (**A**) MCF-7/pBABE (top) and MCF-7/HRG (bottom) cells were transiently co-transfected with an ERE-Luciferase reporter (the ERE-containing reporter plasmid) and pRL/CMV (an internal reporter plasmid to control for transfection efficiency). Cells were incubated for 24 h in the absence or presence of vehicles (control), E_2_ and anti-IL-8 antibody individually or in the combinations specified, and cell extracts were analyzed for Luciferase activity. Data shown represent mean (columns) ± S.D. (bars) (*n* = 3). (**B**) E_2_-independent and E_2_-induced ER transcriptional activity following normalization intra- (top) and inter-normalization (bottom) to the activity of pRL-CMV (1.0-fold). (* *p* < 0.05; ** *p* < 0.005)

**Figure 4 ijms-21-07737-f004:**
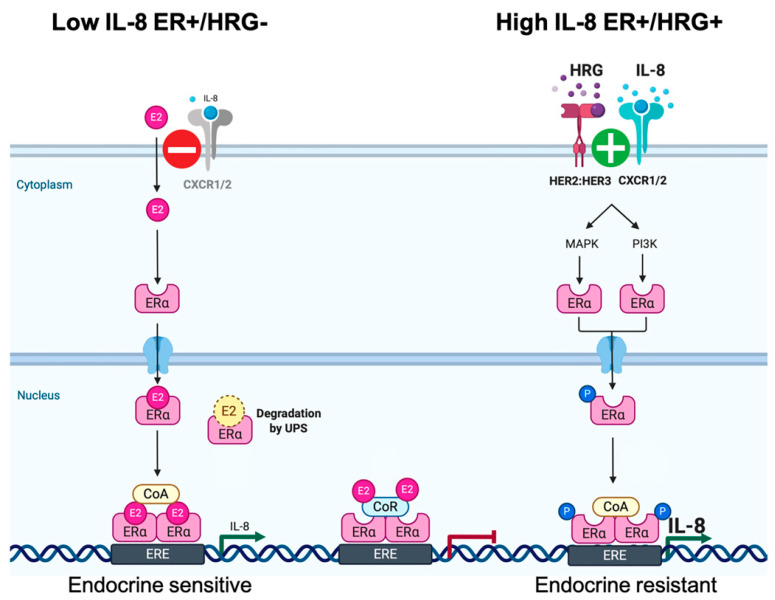
IL-8 participates in the HRG-driven endocrine resistance program in ER+ breast cancer **cells.** The regulatory actions of IL-8 on ER signaling can be related to the HER2:HER3-regulated stage of endocrine responsiveness. In the absence of persistent HER2:HER3 signaling, IL-8 expression might be part of a negative-feedback regulatory mechanism to fine-tune ER signaling. Accordingly, blockade of such negative feedback leads to exacerbated ERα transcriptional activation in response to E_2_. Autocrine HRG-induced heterodimerization and activation of HER2/HER3 stimulate the up-regulation of IL-8 expression and secretion, which in turn might further potentiate the non-genomic (e.g., MAPK- and PI3K-driven) unliganded transcriptional activity of ER characteristic of the endocrine-resistant phenotype in ER+ breast cancer cells.

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
