# Peer review of "Heregulin Drives Endocrine Resistance by Altering IL-8 Expression in ER-Positive Breast Cancer"

_ijms, 2020, doi:10.3390/ijms21207737_

Round 1

Reviewer 1 Report

The current study identified IL-8 as a leading cause of endocrine resistance via human cytokine antibody arrays, providing new knowledge and novel therapeutic implication towards HER2-negative breast cancer. The work can be published after minor revision. 

  1. Figure 1B can be revised as now the first column looks like an outlier (blank between the first two is two big). Similarly, in Figure 3A and B; Another is the symbol behind ER()?  
  2. Have the authors tested the cell viability before and after the blockade of IL-8 upon certain chemotherapeutic? This is also a piece of strong evidence to prove your conclusion. 
  3. A discussion of the limitations and future perspectives (future studies, and applications) should also be included in section Discussion. 
  4. Please revise your conclusion into 3-5 sentences. (the current first 3 sentences are not necessary)

Reviewer 2 Report

The effect of IL-8 action in breast cancer progression was well-described.

  1. Supplementary information, Fig. S1 Cytokines map of the RayBio® Human Cytokine Antibody Array VII & 7.1

-> Presentation of full name of abbreviated cytokines in the footnote is necessary.

2. 2.1. Identification of IL-8 as an HRG-driven cytokine using chemokine antibody array technology

Specifically, MCF-7/pBABE control cells secreted 131 ± 14 pg IL- 8 mg protein-1, whereas MCF-7/HRG, MCF-7/HRG-M4, MCF-7/HRG-M1 cells expressed 440 ± 10, 87 ± 14, and 472 ± 19 pg IL-8 mg-1, respectively.

->Median and interquartile range would be better because of small number of included cells.

  1. 2.2. HRG overexpression in HER2-negative breast cancer cells qualitatively phenocopies the IL-8 cytokine signature driven by HER2 overexpression

Using the antibody-based RayBio™ Human Cytokine Array III, which simultaneously detects 42 cytokines and growth factors.

-> Please provide cytokines of RayBio™ Human Cytokine Array III in the supplementary information

  1. 2.2. HRG overexpression in HER2-negative breast cancer cells qualitatively phenocopies the IL-8 cytokine signature driven by HER2 overexpression

MCF-7/HRG cells, however, MCF-7/Her2-18 cells also showed an elevated secretion of TIMP-2, VEGF and GRO relative to control cells.

-> Please discuss the implication of TIMP-2, VEGF and GRO, which showed difference between MCF-7/HRG, and MCF-7/Her2-18 cells.
